# Multi-label Learning for Large Text Corpora using Latent Variable Model with Provable Guarantees

## Abstract

Here we study the problem of learning labels for large text corpora where each document can be assigned a variable number of labels. The problem is trivial when the label dimensionality is small and can be easily solved by a series of one-vs-all classifiers. However, as the label dimensionality increases, the parameter space of such one-vs-all classifiers becomes extremely large and outstrips the memory. Here we propose a latent variable model to reduce the size of the parameter space, but still efficiently learn the labels. We learn the model using spectral learning and show how to extract the parameters using only three passes through the training dataset. Further, we analyse the sample complexity of our model using PAC learning theory and then demonstrate the performance of our algorithm on several benchmark datasets in comparison with existing algorithms.

## 1 Introduction

Multi-label learning for large text corpora is an upcoming problem in Large-Scale Machine Learning. Unlike the multi-class classification where a document is assigned only one label from a set of labels, here a document can have a variable number of labels. A basic approach to the problem is to use 1-vs-all classification by training a single binary classifier for every label. If the vocabulary size of a text corpus is $D$ and the label dimensionality is $L$, then the 1-vs-all model require $O(DL)$ parameters. Most of the text corpora have moderate to high vocabulary size ($D$), and 1-vs-all classification is feasible as long as $D \gg L$. However, as the number of labels increases to a point when $L \sim D$, the size of the parameter space increases to $O(D^2)$, and we can no longer store the 1-vs-all models in the memory Yu et al. (2014).

A fairly intuitive approach to reducing the size of the parameter space is to use low-rank models. The existing low-rank approaches in the literature are mainly based on discriminative models that use a mapping $\Phi : \mathbb{R}^D \to \mathbb{R}^L$ between the word space and the label space. If the rank of the model is restricted to $K \ll D$, then such mappings have the form $Z = HW^\top$, where $W \in \mathbb{R}^{D \times K}$ and $H \in \mathbb{R}^{L \times K}$ contain the low-rank features of the words and the labels respectively. The parameter space of such models has the size $\Theta\left((D + L)K\right)$, which is much smaller than $O(DL)$ for large values of $L$ and can be conveniently stored the memory.

Amongst the recent literature on discriminative low-rank models, WSABIE Weston et al. (2011) defines weighted approximate pair-wise rank (WARP) loss on the mapping and optimises the loss on the training dataset. LEML Yu et al. (2014) generalises the loss function to squared-loss, sigmoid loss or hinge loss, which are typical of Linear Regression, Logistic Regression, and Linear SVM respectively. Both of LEML and WSABIE use gradient descent to optimise the loss on training datasets and therefore are susceptible to overfitting. Also, both of them run several iterations through the dataset and can slow down for large datasets containing millions of text instances.

Here we propose a latent variable model for multi-label learning. Our model can be viewed as a generative counterpart of the discriminative low-rank models like WSABIE or LEML. Unlike the usual cases where such latent variable models are trained using EM algorithm, we use Spectral Learning or Method of Moments Anandkumar et al. (2014) to extract the parameters. Method of Moments is an upcoming non-iterative approach for latent variable based models, and has successfully been applied in Spherical Gaussian Model Hsu & Kakade (2013), Hidden Markov Models

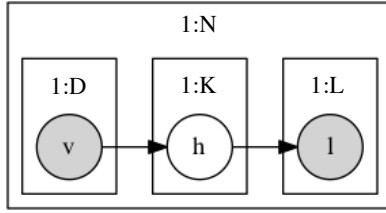

Figure 1: Latent Variable Model

(Hsu et al. (2012) and Song et al. (2010)), Bayesian Non-parametrics Tung & Smola (2014), Topic Models Anandkumar et al. (2012), and in various NLP applications (Dhillon et al. (2012) and Cohen et al. (2014)). We first derive a closed form expression to estimate the model parameters using the Method of Moments. Further, we derive the convergence bounds on the estimated parameters, and then compare the performance of our algorithm with the discriminative low-rank models both from the perspective of accuracy and computation time.

There are other approaches beyond low-rank mappings, such as FastXML Prabhu & Varma (2014) that uses tree-based hierarchies for the labels, or SLEEC Bhatia et al. (2015) that uses kNN based embedding of the texts in the training dataset. However, these approaches use a combination of different algorithms and are geared towards end-to-end solutions. Whereas, we look into the problem from a modelling perspective and limit our discussion to the low-rank models since they have a similar model complexity to the latent variable model.

## 2 METHODOLOGY

We build our model based on the "bag of words" assumption on the documents taking into account only the occurrence of the words, not their counts. Let us assume that there are $N$ labeled documents in a corpus with vocabulary size $D$ and label dimensionality $L$. Then using a latent variable $h$ that assumes $K$ states, the probability of a label $l$ being assigned to a document $d$ can be expressed as:

$$P[l|d] = \sum_{k=1}^{K} P[l|h = k]P[h = k|d] \tag{1}$$

where the conditional probability of a document $d$ given $h$ can be modelled from the set of distinct words $\mathcal{W}_d$ present in it as:

$$P[d|h = k] = \prod_{v \in \mathcal{W}_d} P[v|h = k] \prod_{v \notin \mathcal{W}_d} \left(1 - P[v|h = k]\right) \tag{2}$$

From here, $P[h = k|d]$ can be expressed using Bayes Theorem as:

$$P[h = k|d] = \frac{P[h = k]P[d|h = k]}{\sum_{k=1}^{K} P[h = k]P[d|h = k]} \tag{3}$$

The latent variable model is described by the plate notation in Figure 1, where the words are denoted by $v$, the labels by $l$ and the latent variable by $h$. Our model parameters are $P[h]$, $P[v|h]$ and $P[l|h]$, and the number of parameters is $\Theta\left((D + L)K\right)$, same as the discriminative low-rank models.

Let us define $\pi_k \in [0, 1]$ as the probability of the latent variable $h$ assuming the state $k \in [K]$.

$$\pi_k = P[h = k] \tag{4}$$

Let us define $\mu_k \in [0, 1]^D$ as the probability vector of a word $v$ conditional to the latent state $k \in [K]$.

$$\mu_k = P[v|h = k] \tag{5}$$

Similarly, let us define $\gamma_k \in [0,1]^L$ as the probability vector of a label $l$ conditional to the latent state $k \in [K]$.

$$\gamma_k = P[l|h = k] \tag{6}$$

One key difference of our parameterization from Anandkumar et al. (2012) and Wang & Zhu (2014) is that they define $\{\mu_k\}_{k=1}^K$ as the expected count of the words ($\mathbb{E}[v|h = k]$) for each topic $k$, while we define $\{\mu_k\}_{k=1}^K$ as the probability of the words. A probabilistic formulation enables us to model the documents using Equation 2, which is not possible using expectations. We later use the document probabilities to predict the labels for a test document (Section 4). Therefore, unlike Anandkumar et al. (2012) or Wang & Zhu (2014) that uses moments on the word counts defined on the domain $\bigotimes^p \mathbb{R}^D$ (for $p$th order moment), we use moments based on the joint probability of the words defined on the domain $\bigotimes^p [0,1]^D$. We outline the derivation for the probability moments in this section, and then show that our formulation achieves a tighter convergence bound on the parameters in Section 3.1. The label parameters $\{\gamma_k\}_{k=1}^K$ are unique to our model.

## 2.1 MOMENT FORMULATION

We first try to formulate the joint probability of a pair of words in terms of $\pi$ and $\mu$. Let us assume that we choose two words $w_1$ and $w_2$ from a document at random. The term $P[w_1 = v_i, w_2 = v_j]$ represents the probability by which any pair of words picked at random from a document turns out to be $v_i$ and $v_j$, and it is nothing but $P[v_i, v_j]$, where $i, j \in [D]$. Similarly, $P[w_1 = v_i]$ is same as $P[v_i]$, and the same holds for $P[w_2 = v_j]$.

Following the generative model in Figure 1, any two words $w_1$ and $w_2$ in a document are conditionally independent given $h$. Therefore,

$$P[w_1 = v_i, w_2 = v_j]$$
$$= \sum_{k=1}^K P[w_1 = v_i, w_2 = v_j|h = k]P[h = k]$$
$$= \sum_{k=1}^K P[w_1 = v_i|h = k]P[w_2 = v_j|h = k]P[h = k]$$
$$= \sum_{k=1}^K P[v_i|h = k]P[v_j|h = k]P[h = k]$$
$$= \sum_{k=1}^K \pi_k \mu_{ki} \mu_{kj}$$

where $\mu_{ki}$ is the $i$th element of the vector $\mu_k$ and so on.

$$\therefore P[v_i, v_j] = \sum_{k=1}^K \pi_k \mu_{ki} \mu_{kj} \quad \forall i, j \in [D] \tag{7}$$

If $M_2 \in [0,1]^{D \times D}$ is the matrix containing the joint probability of the pairs of words with $M_{2_{i,j}} = P[v_i, v_j] \, \forall i, j \in [D]$, then it can be expressed as:

$$M_2 = \sum_{k=1}^K \pi_k \mu_k \mu_k^\top \tag{8}$$

Similarly, if the tensor $M_3 \in [0,1]^{D \times D \times D}$ is defined as the moment containing the joint probability of a triplet of words with $M_{3_{i,j,\kappa}} = P[v_i, v_j, v_\kappa], \forall i, j, \kappa \in [D]$, then

$$M_3 = \sum_{k=1}^K \pi_k \mu_k \otimes \mu_k \otimes \mu_k \tag{9}$$

where $\otimes$ stands for the tensor product.

Finally, we define $M_L \in [0,1]^{L \times D \times D}$ as the probability moment of a label occurring with a pair of words such that $M_{L_{\kappa,i,j}} = P[l_\kappa, v_i, v_j], \forall \kappa \in [L]$ and $\forall i, j \in [D]$. Then,

$$M_L = \sum_{k=1}^{K} \pi_k \gamma_k \otimes \mu_k \otimes \mu_k \tag{10}$$

## 2.2 Closed Form Expression of the Parameters

We derive the equations for extracting $\pi$, $\mu$ and $\gamma$ in this section. The first step is to whiten the matrix $M_2$, where we try to find a matrix low rank $W$ such that $W^\top M_2 W = I$. This is similar to the whitening in ICA Hyvärinen et al. (2004), with the covariance matrix being replaced by the co-occurrence probability matrix. The whitening is usually done through singular value decomposition of $M_2$. If the $K$ maximum singular values of $M_2$ are $\{\nu_k\}_{k=1}^{K}$, and $\{\omega_k\}_{k=1}^{K}$ are the corresponding left singular vectors, then the whitening matrix of rank $K$ is computed as $W = \Omega \Sigma^{-1/2}$, where $\Omega = [\omega_1 | \omega_2 | \dots | \omega_K]$ and $\Sigma = \mathrm{diag}(\nu_1, \nu_2, \dots, \nu_K)$.

Upon whitening $M_2$ takes the form

$$W^\top M_2 W = W^\top \Big( \sum_{k=1}^{K} \pi_k \mu_k \mu_k^\top \Big) W = \sum_{k=1}^{K} \big( \sqrt{\pi_k} W^\top \mu_k \big) \big( \sqrt{\pi_k} W^\top \mu_k \big)^\top = \sum_{k=1}^{K} \tilde{\mu}_k \tilde{\mu}_k^\top = I \tag{11}$$

Hence $\tilde{\mu}_k = \sqrt{\pi_k} W^\top \mu_k$ are orthonormal vectors. Multiplying $M_3$ along all three dimensions by $W$, we get

$$\tilde{M}_3 = M_3(W, W, W) = \sum_{k=1}^{K} \pi_k (W^\top \mu_k) \otimes (W^\top \mu_k) \otimes (W^\top \mu_k) = \sum_{k=1}^{K} \frac{1}{\sqrt{\pi_k}} \tilde{\mu}_k \otimes \tilde{\mu}_k \otimes \tilde{\mu}_k \tag{12}$$

$\tilde{M}_3$ is a tensor in the domain $\mathbb{R}^{K \times K \times K}$. Upon the factorisation of $\tilde{M}_3$, if the eigenvalues and eigenvectors are $\{\lambda_k\}_{k=1}^{K}$ and $\{u_k\}_{k=1}^{K}$ respectively, then $\lambda_k = \frac{1}{\sqrt{\pi_k}} \implies \pi_k = \lambda_k^{-2}$, and

$$u_k = \tilde{\mu}_k = \sqrt{\pi_k} W^\top \mu_k = \frac{1}{\lambda_k} W^\top \mu_k \tag{13}$$

Hence, $\{\mu_k\}_{k=1}^{K}$ can be recovered as $\mu_k = \lambda_k W^\dagger u_k$, where $W^\dagger = W (W^\top W)^{-1}$ is the pseudo-inverse of $W^\top$. Since we compute $P[v|h]$ by normalising $\mu_k$ as $P[v|h=k] = \frac{\mu_{kv}}{\sum_v \mu_{kv}}$, it suffices to compute $\mu_k = W^\dagger u_k$ as $\lambda_k$ will be cancelled during normalisation.

It is possible to compute $\gamma$ through the factorisation of second and third order moments of the labels in a similar way to $\mu$. However, there is no guarantee that the label probabilities of $\gamma_k$ will correspond to the same topic as the word probabilities in $\mu_k$ for every $k$ when we compute them separately. Without the topic alignment between $\mu$ and $\gamma$, label prediction is not possible.

To overcome this limitation, we use the cross-moment $M_L$ to compute $\gamma$. If we multiply $M_L$ twice by $W$, then

$$M_L(W, W) = \sum_{k=1}^{K} \pi_k \gamma_k \otimes (W^\top \mu_k) \otimes (W^\top \mu_k) = \sum_{k=1}^{K} \gamma_k \otimes (\sqrt{\pi_k} W^\top \mu_k) \otimes (\sqrt{\pi_k} W^\top \mu_k)$$

$$= \sum_{k=1}^{K} \gamma_k \otimes \tilde{\mu}_k \otimes \tilde{\mu}_k \tag{14}$$

If $u_k$ is the $k$th eigenvector of $\tilde{M}_3$, then

$$u_k^\top M_L(W, W) u_k = \tilde{\mu}_k^\top M_L(W, W) \tilde{\mu}_k = \tilde{\mu}_k^\top \left( \sum_{k=1}^{K} \gamma_k \otimes \tilde{\mu}_k \otimes \tilde{\mu}_k \right) \tilde{\mu}_k = \gamma_k \tag{15}$$

since $\tilde{\mu}_k$s are orthonormal. Computing $\gamma_k$ by this method ensures that $\gamma_k$ and $\tilde{\mu}_k$ correspond to the same topic for every $k$, which in turn ensures that $\gamma_k$ and $\mu_k$ correspond to the same topic.

## 3 PARAMETER ESTIMATION

So far we have shown how to extract the parameters from the global moments $M_2$, $M_3$ and $M_L$ defined on population. In practice, we cannot compute the population parameters; all we can do is to estimate them from the sample in hand and compute an error bound for the estimation. We denote the estimated values of $M_2$, $M_3$, $M_L$, $\pi$, $\mu$, $\gamma$, $W$ and $U$ from the sample as $\hat{M}_2$, $\hat{M}_3$, $\hat{M}_L$, $\hat{\pi}$, $\hat{\mu}$, $\hat{\gamma}$, $\hat{W}$ and $\hat{U}$, conforming with the notations used in the previous literature.

If we take into account only the occurrence of each distinct word in a document, then using "one hot encoding," we can represent the words as a binary vector $x \in \{0,1\}^D$, and the labels as another $y \in \{0,1\}^L$. If there are $N$ samples, then we can represent the words in the entire corpus as $X \in \{0,1\}^{N \times D}$, and the associated labels as $Y \in \{0,1\}^{N \times L}$. The pairwise counts of the words can be estimated by $X^\top X$, whose sum of all elements is,

$$\sum_{v_1=1}^{D} \sum_{v_2=1}^{D} (X^\top X)_{v_1,v_2} = \sum_{v_1=1}^{D} \sum_{v_2=1}^{D} \sum_{i=1}^{N} x_{i,v_1} x_{i,v_2} = \sum_{i=1}^{N} \sum_{v_1=1}^{D} \sum_{v_2=1}^{D} x_{i,v_1} x_{i,v_2} = \sum_{i=1}^{N} nnz(x_i)^2$$

where $x_i$ is the row of $X$ corresponding to the $i$th document, $x_{i,v}$ is the $v$th element in it ($v \in [D]$), and $nnz(x_i)$ is the number of non-zero elements in $x_i$, i.e., the number of distinct words in the $i$th document.

Therefore, the joint probability matrix of the words ($M_2$) can be estimated as,

$$\hat{M}_2 = \frac{1}{\sum_{i=1}^{N} nnz(x_i)^2} X^\top X \tag{16}$$

Similarly, the triple-wise occurrences of the words can be estimated by the tensor $X \otimes X \otimes X$, and the sum of all of the elements of this tensor is

$$\sum_{v_1=1}^{D} \sum_{v_2=1}^{D} \sum_{v_3=1}^{D} (X \otimes X \otimes X)_{v_1,v_2,v_3} = \sum_{i=1}^{N} nnz(x_i)^3$$

Therefore, $M_3$ can be estimated as

$$\hat{M}_3 = \frac{1}{\sum_{i=1}^{N} nnz(x_i)^3} X \otimes X \otimes X \tag{17}$$

The dimensions of $\hat{M}_2$ and $\hat{M}_3$ are $D^2$ and $D^3$ respectively. But in practice, these quantities are extremely sparse. We do not need to compute $\hat{M}_3$ in practice. If the whitening matrix computed through the Singular Value Decomposition of $\hat{M}_2$ is $\hat{W}$, then $\tilde{M}_3$ can be estimated straight away using Equation 12 as,

$$\hat{\tilde{M}}_3 = \frac{1}{\sum_{i=1}^{N} nnz(x_i)^3} X\hat{W} \otimes X\hat{W} \otimes X\hat{W} \tag{18}$$

Since $\hat{\tilde{M}}_3$ has a dimension of $K^3$ and $K \ll D$, it can be conveniently stored in the memory. The computation of $\hat{M}_2$ takes one pass through the dataset, and the computation of $\hat{\tilde{M}}_3$ takes another.

Also, the counts of the labels occurring with the pairs of words can be estimated by the tensor $Y \otimes X \otimes X$, whose sum of all element is

$$\sum_{l=1}^{L} \sum_{v_1=1}^{D} \sum_{v_2=1}^{D} (Y \otimes X \otimes X)_{l,v_1,v_2} = \sum_{i=1}^{N} nnz(y_i) nnz(x_i)^2$$

where $y_i$ represents the $i$th row of $Y$ and $nnz(y_i)$ is the number of labels associated with the $i$th document. Therefore, $M_L$ can be estimated as,

$$\hat{M}_L = \frac{1}{\sum_{i=1}^{N} nnz(y_i) nnz(x_i)^2} Y \otimes X \otimes X \tag{19}$$

---

**Algorithm 1** Three-pass Algorithm for Parameter Extraction

---

**Input:** Sparse Binary Data $X \in \{0,1\}^{N \times D}$, Labels $Y \in \{0,1\}^{N \times L}$ and $K \in \mathbb{Z}^+$
**Output:** $\hat{\pi}$, $\hat{P}[v|h]$ and $\hat{P}[l|h]$

1. Estimate $\hat{M}_2 = (X^\top X)/\sum_{i=1}^{N} nnz(x_i)^2$              (pass # 1)
2. Compute $K$ maximum singular values of $\hat{M}_2$ as $\{\nu_k\}_{k=1}^{K}$, and corresponding left singular vectors as $\{\omega_k\}_{k=1}^{K}$. Define $\Omega = [\omega_1|\omega_2|\dots|\omega_K]$, and $\Sigma = \mathrm{diag}(\nu_1, \nu_2, \dots, \nu_K)$
3. Estimate the whitening matrix $\hat{W} = \Omega \Sigma^{-1/2} \in \mathbb{R}^{D \times K}$
4. Estimate $\hat{\tilde{M}}_3 = (X\hat{W} \otimes X\hat{W} \otimes X\hat{W})/\sum_{i=1}^{N} nnz(x_i)^3$      (pass # 2)
5. Compute eigenvalues $\{\hat{\lambda}_k\}_{k=1}^{K}$ and eigenvectors $\{\hat{u}_k\}_{k=1}^{K}$ of $\hat{\tilde{M}}_3$. Assign $\hat{U} = [\hat{u}_1|\hat{u}_2|\dots|\hat{u}_K]$
6. Estimate $\hat{\mu}_k = \hat{W}^\dagger \hat{u}_k$, where $\hat{W}^\dagger = \hat{W}(\hat{W}^\top \hat{W})^{-1}$, and $\hat{\pi}_k = \hat{\lambda}_k^{-2}$, $\forall k \in 1, 2 \dots K$.
7. Estimate
$$\hat{P}[v|h = k] = \frac{\hat{\mu}_{kv}}{\sum_v \hat{\mu}_{kv}}, \forall k \in 1 \dots K, v \in v_1 \dots v_D$$
8. Estimate $\hat{\Gamma} = Y^\top ((X\hat{W}\hat{U}) \bullet (X\hat{W}\hat{U}))$, where $\bullet$ stands for the element-wise product (pass # 3)
9. Estimate
$$\hat{P}[l|h = k] = \frac{\hat{\Gamma}_{lk}}{\sum_l \hat{\Gamma}_{lk}}, \forall k \in 1 \dots K, l \in l_1 \dots l_L$$

---

If $k$th eigenvector of $\hat{\tilde{M}}_3$ is $\hat{u}_k$, then from Equation 15

$$\begin{aligned}
\hat{\gamma}_k &= \hat{u}_k^\top \hat{M}_L(\hat{W}, \hat{W})\hat{u}_k \\
&= \frac{1}{\sum_{i=1}^{N} nnz(y_i)nnz(x_i)^2} \hat{u}_k^\top (Y \otimes X\hat{W} \otimes X\hat{W})\hat{u}_k \\
&= \frac{1}{\mathcal{Z}(X,Y)} Y \otimes X\hat{W}\hat{u}_k \otimes X\hat{W}\hat{u}_k \\
&= \frac{1}{\mathcal{Z}(X,Y)} Y^\top ((X\hat{W}\hat{u}_k) \bullet (X\hat{W}\hat{u}_k))
\end{aligned}$$

where $\bullet$ stands for the element-wise product, and $\mathcal{Z}(X,Y) = \sum_{i=1}^{N} nnz(y_i)nnz(x_i)^2$ is a constant. If we assign $\hat{\Gamma} = [\hat{\gamma}_1|\hat{\gamma}_2|\dots|\hat{\gamma}_K]$, then it can be computed as,

$$\begin{aligned}
\hat{\Gamma} &= [\hat{\gamma}_1|\hat{\gamma}_2|\dots|\hat{\gamma}_K] \\
&= \frac{1}{\mathcal{Z}(X,Y)} Y^\top \Big[ (X\hat{W}\hat{u}_1 \bullet X\hat{W}\hat{u}_1)|(X\hat{W}\hat{u}_2 \bullet X\hat{W}\hat{u}_2)| \\
&\qquad\qquad\qquad\qquad \dots |(X\hat{W}\hat{u}_K \bullet X\hat{W}\hat{u}_K) \Big] \\
&= \frac{1}{\mathcal{Z}(X,Y)} Y^\top ((X\hat{W}\hat{U}) \bullet (X\hat{W}\hat{U})) \qquad\qquad\qquad (20)
\end{aligned}$$

where $\hat{U} = [\hat{u}_1|\hat{u}_2|\dots|\hat{u}_K]$. Since we estimate $P[l|h]$ by normalising the columns of $\hat{\Gamma}$ as $\hat{P}[l|h = k] = \frac{\hat{\Gamma}_{lk}}{\sum_l \hat{\Gamma}_{lk}}$, it is sufficient to compute $\hat{\Gamma} = Y^\top ((X\hat{W}\hat{U}) \bullet (X\hat{W}\hat{U}))$ as the constant $\mathcal{Z}(X,Y)$ will be cancelled out during normalisation. Instead of computing all the $\hat{\gamma}_k$s separately, we can compute $\hat{\Gamma}$ in one step by just one pass through the entire dataset. The overall method takes three passes through the dataset as outlined in Algorithm 1. There is no need to compute $\hat{M}_L$ explicitly since $\hat{\Gamma}$ can be computed straight away from $X$, $Y$ and the intermediate parameters.

### 3.1 CONVERGENCE BOUNDS

**Theorem 1.** *Let us assume that we run Algorithm 1 on $N$ i.i.d. samples with word vectors $x_1, x_2 \ldots x_N$ and label vectors $y_1, y_2 \ldots y_N$. Let us define $\varepsilon_1 = \left(1 + \sqrt{\frac{\log(1/\delta)}{2}}\right)$ and $\varepsilon_2 = \left(1 + \sqrt{\frac{\log(2/\delta)}{2}}\right)$ for some $\delta \in (0,1)$. If $N \geq \max(n_1, n_2, n_3)$, where*

1. $n_1 = c_2 \left(\log K + \log \log \left(\frac{K}{c_1} \cdot \sqrt{\frac{\pi_{max}}{\pi_{min}}}\right)\right)$

2. $n_2 = \Omega\left(\left(\frac{\varepsilon_1}{\tilde{d}_{2s}\sigma_K(M_2)}\right)^2\right)$

3. $n_3 = \Omega\left(K^2 \left(\frac{10}{\tilde{d}_{2s}\sigma_K(M_2)^{5/2}} + \frac{2\sqrt{2}}{\tilde{d}_{3s}\sigma_K(M_2)^{3/2}}\right)^2 \varepsilon_1^2\right)$

*for some constants $c_1$ and $c_2$, then following bounds on the estimated parameters hold with probability at least $1 - \delta$,*

$$\|\mu_k - \hat{\mu}_k\| \leq \left(\frac{160\sqrt{\sigma_1(M_2)}}{\tilde{d}_{2s}\sigma_K(M_2)^{5/2}} + \frac{32\sqrt{2\sigma_1(M_2)}}{\tilde{d}_{3s}\sigma_K(M_2)^{3/2}} + \frac{4\sqrt{\sigma_1(M_2)}}{\tilde{d}_{2s}\sigma_K(M_2)}\right) \frac{\varepsilon_1}{\sqrt{N}}$$

$$\|\gamma_k - \hat{\gamma}_k\| \leq \left(\frac{160}{\tilde{d}_{2s}\sigma_K(M_2)^{7/2}} + \frac{32\sqrt{2}}{\tilde{d}_{3s}\sigma_K(M_2)^{5/2}} + \frac{2+\sqrt{2}}{\tilde{d}_{2s}\sigma_K(M_2)^2}\right) \frac{2\varepsilon_1}{\sqrt{N}} + \frac{8\varepsilon_2}{\tilde{d}_{ls}\sigma_K(M_2)\sqrt{N}}$$

$$|\pi_k - \hat{\pi}_k| \leq \left(\frac{200}{\sigma_K(M_2)^{5/2}} + \frac{40\sqrt{2}}{\sigma_K(M_2)^{3/2}}\right) \frac{\varepsilon_1}{\tilde{d}_{3s}\sqrt{N}}$$

*The terms $\sigma_1(M_2) \ldots \sigma_K(M_2)$ are the $K$ largest eigenvalues of the matrix $M_2$, whereas $\tilde{d}_{2s} = \mathbb{E}\left[nnz(x)^2\right]$, $\tilde{d}_{3s} = \mathbb{E}\left[nnz(x)^3\right]$ and $\tilde{d}_{ls} = \mathbb{E}\left[nnz(y)nnz(x)^2\right]$, with $nnz(x)$ representing the number of distinct words and $nnz(y)$ representing the number of labels present in an instance. The proof is included in the appendix.*

The convergence bounds on $\pi$ and $\mu$ are very similar to those in Anandkumar et al. (2012) and Wang & Zhu (2014), except for the terms $\tilde{d}_{2s}$, $\tilde{d}_{3s}$ and $\tilde{d}_{ls}$ in the denominator. These terms arise in our convergence bounds since we use the probability moments, whereas Anandkumar et al. (2012) and Wang & Zhu (2014) use the moments of word counts. The parameter $\gamma$ is unique to our model, although its bound contains those terms too.

We need at least one document with 3 distinct words or more to construct the third order moment $M_3$, i.e., $nnz(x) \geq 3$ for at least one document. Therefore, $\tilde{d}_{2s} = \mathbb{E}\left[nnz(x)^2\right] > 1$ and $\tilde{d}_{3s} = \mathbb{E}\left[nnz(x)^3\right] > 1$. Also, since every document has least one label, $nnz(y) \geq 1$ for all the documents, and therefore, and $\tilde{d}_{ls} > 1$. In practice, these terms can be much larger than 1 for any real-life text corpus (The estimates of them for different experimental datasets are in Table 1). Therefore, our algorithm achieves a tighter convergence bound than Anandkumar et al. (2012) or Wang & Zhu (2014). Further, since the terms $\tilde{d}_{2s}$ and $\tilde{d}_{3s}$ appear in the denominators of $n_2$ and $n_3$ too, using probability moments also lowers the minimum number of samples required.

### 3.2 COMPUTATIONAL COMPLEXITY

The bottleneck of the algorithm is the whitening step of $\hat{M}_2$, especially for the large datasets. $\hat{M}_2$ computed using Equation 16 is a symmetric p.s.d. matrix with very high sparsity. The number of elements of $\hat{M}_2$ is $O\left(\sum_{i=1}^{N} nnz(x_i)^2\right)$, with the worst case occurring when no two documents have any word in common, and every element in $X^\top X$ is 1. The top $K$ eigenvalues for large symmetric matrices are usually computed through Lanczos method. For a sparse matrix of size $D \times D$ with $O\left(\sum_{i=1}^{N} nnz(x_i)^2\right)$ number of elements, this step has a complexity of $O\left(\left(\sum_{i=1}^{N} nnz(x_i)^2\right)K + DK^2\right)$ Mahadevan (2008).

We do not compute the third order tensors $\hat{M}_3$ or $\hat{M}_L$ explicitly. The step to compute $\hat{\tilde{M}}_3$ using Equation 18 has a complexity of $O(NK^3)$, while the computation of $\hat{\Gamma}_L$ using Equation 20 has a complexity of $O\big(\big(\sum_{i=1}^{N} nnz(y_i)\big)K^2\big)$. We carry out the tensor factorisation of $\hat{\tilde{M}}_3 \in \mathbb{R}^{K \times K \times K}$ using the Tensor Toolbox Bader et al. (2015), and this step has a complexity of $O(K^4 \log \epsilon)$ to compute each of the $K$ eigenvalues up to an accuracy of $\epsilon$ Kolda & Mayo (2011). These steps contribute the most to the computational cost. The overall time complexity of Algorithm 1 is $O\big(\big(\sum_{i=1}^{N} nnz(x_i)^2\big)K + \big(D + \sum_{i=1}^{N} nnz(y_i)\big)K^2 + NK^3 + K^4 \log \epsilon\big)$.

As per as space requirement is concerned, the storage of $\hat{M}_2$ takes up $O\big(\sum_{i=1}^{N} nnz(x_i)^2\big)$ space, whereas the word parameters like $\hat{W}$ and $\hat{\mu}_{1:K}$ take up a space of $\Theta(DK)$, the label parameter $\hat{\Gamma}$ takes up a space of $\Theta(LK)$, and $\hat{\tilde{M}}_3$ takes up a space of $\Theta(K^3)$. The overall space complexity is $O\big(\sum_{i=1}^{N} nnz(x_i)^2 + (D + L)K + K^3\big)$.

## 4 LABEL PREDICTION

Once we have extracted the model parameters $\hat{P}[h]$, $\hat{P}[v|h]$ and $\hat{P}[l|h]$ using Algorithm 1, a new test document $d$ with a distinct set of words $\mathcal{W}_d$ can be modelled as,

$$\hat{P}[d|h=k] = \prod_{v \in \mathcal{W}_d} \hat{P}[v|h=k] \prod_{v \notin \mathcal{W}_d} \left(1 - \hat{P}[v|h=k]\right)$$

This step will take $D$ multiplications involving floating point operations. However, since $\sum_v \hat{P}[v|h=k] = 1$ for $v \in \{v_1 \ldots v_D\}$ and $D$ can be very large, the values of $\hat{P}[v|h=k]$ are much less than one for the most, and $\left(1 - \hat{P}[v|h=k]\right) \approx 1$. If we threshold $\hat{P}[v|h=k]$ when $v \notin \mathcal{W}_d$ using some threshold $\eta$ (say $\eta = 0.01$ or $0.001$), then $\hat{P}[d|h=k]$ takes the form:

$$\hat{P}[d|h=k] = \prod_{v \in \mathcal{W}_d} \hat{P}[v|h=k] \prod_{\substack{v \notin \mathcal{W}_d \\ \hat{P}[v|h=k]>\eta}} \left(1 - \hat{P}[v|h=k]\right)$$

Usually only a few of $\hat{P}[v|h=k]$ will be greater than the threshold, and this will significantly reduce the number of multiplications when $v \notin \mathcal{W}_d$ without sacrificing accuracy. Also, since a document has only a few distinct words, i.e. $|\mathcal{W}_d| \ll D$, the left multiplicand (when $v \in \mathcal{W}_d$) accounts for only a few multiplications. From there, we can estimate the following using Bayes rule.

$$\hat{P}[h=k|d] = \frac{\hat{P}[h=k]\hat{P}[d|h=k]}{\sum_{k=1}^{K} \hat{P}[h=k]\hat{P}[d|h=k]} = \frac{\hat{\pi}_k \prod_{v \in \mathcal{W}_d} \hat{P}[v|h=k] \prod_{\substack{v \notin \mathcal{W}_d \\ \hat{P}[v|h=k]>\eta}} \left(1 - \hat{P}[v|h=k]\right)}{\sum_{k=1}^{K} \hat{\pi}_k \prod_{v \in \mathcal{W}_d} \hat{P}[v|h=k] \prod_{\substack{v \notin \mathcal{W}_d \\ \hat{P}[v|h=k]>\eta}} \left(1 - \hat{P}[v|h=k]\right)}$$

And finally, the label probabilities for the document ($\hat{P}[l|d]$) can be estimated from $\hat{P}[l|h]$ and $\hat{P}[h|d]$ using Equation 1.

## 5 EXPERIMENTAL RESULTS

We carry out our experiments on six datasets ranging from small datasets like Bibtex with $4,880$ training instances to large datasets like WikiLSHTC with more than 1.7M training instances. The description of the datasets are listed in Table 1. We categorise the datasets into three groups:
1. Small: Bibtex and Delicious
2. Medium: Wiki-31K and NYTimes
3. Large: AmazonCat and WikiLSHTC

| Name | # of Train Points | # of Test Points | Feature Dimension | Label Dimension | Average # of Features | Median # of Features | Average # of Labels | Median # of Labels | $\hat{\bar{d}}_{2s}$ | $\hat{\bar{d}}_{3s}$ | $\hat{\bar{d}}_{ls}$ |
|---|---|---|---|---|---|---|---|---|---|---|---|
| Bibtex | 4,880 | 2,515 | 1,836 | 159 | 68.67 | 69 | 2.4 | 2 | 5,957 | 597K | 15K |
| Delicious | 12,920 | 3,185 | 500 | 983 | 18.29 | 6 | 19.02 | 20 | 1,892 | 363K | 24K |
| Wiki-31K | 14,146 | 6,616 | 101,938 | 30,938 | 669.05 | 513 | 18.64 | 19 | 559K | 545M | 11M |
| NYTimes | 14,669 | 15,989 | 24,670 | 4,185 | 373.91 | 354 | 5.40 | 5 | 175K | 95M | 755K |
| AmazonCat | 1,186,239 | 306,782 | 203,882 | 13,330 | 71.09 | 45 | 5.04 | 4 | 6,301 | 651K | 27K |
| WikiLSHTC | 1,778,351 | 587,084 | 1,617,899 | 325,056 | 42.15 | 30 | 3.19 | 2 | 2,210 | 135K | 7,594 |

Table 1: Description of the Datasets

| True Labels | LEML | MoM |
|---|---|---|
| "airlines and airplanes", "hijacking", "terrorism" | **"airlines and airplanes"** (0.34), **"terrorism"** (0.30), "united states international relations" (0.27), "elections" (0.22), "armament, defense and military forces" (0.18), "internationalrelations" (0.18), "bombs and explosives" (0.15), "murders and attempted murders " (0.13), "biographical information" (0.13), "islam" (0.12) | **"terrorism"** (0.12), "united states international relations" (0.08), **"airlines and airplanes"** (0.07), "world trade center (nyc)" (0.07), **"hijacking"** (0.07), "united states armament and defense" (0.07), "pentagon building" (0.03), "bombs and explosives" (0.03), "islam" (0.02), "missing persons " (0.02) |
| "armament, defense and military forces", "civil war and guerrilla warfare", "politics and government" | **"civil war and guerrilla warfare"** (0.62), "united states international relations" (0.39), "united states armament and defense" (0.23), **"armament, defense and military forces"** (0.23), "internationalrelations" (0.17), "oil (petroleum) and gasoline" (0.11), "surveys and series" (0.10), "military action" (0.09), "foreign aid" (0.08), "independence movements" (0.08) | **"civil war and guerrilla warfare"** (0.09), "united states international relations" (0.09), "united states armament and defense" (0.06), **"politics and government"** (0.04), **"armament, defense and military forces"** (0.03), "internationalrelations" (0.02), "immigration and refugees" (0.02), "foreign aid" (0.02), "terrorism" (0.02), "economic conditions and trends" (0.02) |

Table 2: Examples of label prediction from the NYTimes dataset. The numbers in parenthesis are the scores for the top 10 labels. The scores of LEML and MoM have different ranges.

Since LEML is shown to outperform WSABIE and other benchmark algorithms on various small and large-scale datasets in Yu et al. (2014), we benchmark the performance of our method against LEML. Also, there are other methods proposed on topic based embedding of the labels, most recent of which is Rai et al. (2015) that extends Latent Dirichlet Allocation Blei et al. (2003) to Multi-label learning, and uses Bayesian Learning through Gibbs Sampling. However, use of Gibbs sampler limits the use of the algorithms only to the datasets of limited size. The largest dataset used in Rai et al. (2015) is EurLex that is similar to NYTimes dataset in size. Using any MCMC based sampling scheme is not viable for the large datasets such as AmazonCat or WikiLSHTC containing millions of training instances.

In our experiments, we chose to measure AUC (of Receiver Operator Characteristics) against the latent dimensionality ($K$). AUC is a versatile measure and has been used in a range of problems from binary classification to ranking. Also, it is shown that there exists a one-to-one relation between AUC and Precision-Recall curves in Davis & Goadrich (2006), i.e., the same algorithm achieving a higher AUC will also produce a better Precision-Recall curve. We carried out our experiments on Unix Platform on a single machine with Intel i5 Processor (2.4GHz) and 16GB memory, and no multi-threading or any other performance enhancement method is used in the code.[1] For the label prediction step of MoM, we chose $\eta = 0.001$. For LEML, we ran ten iterations for the small datasets and five iterations for the medium and large datasets, since the authors of LEML chose a similar number of iterations for their experiments in Yu et al. (2014). For AmazonCat and WikiLSHTC datasets, we ran LEML on an i2.4xlarge instance of Amazon EC2 with 122 GB of memory since LEML needs significantly larger memory for these two datasets (Figure 2).

---

[1]To be shared later

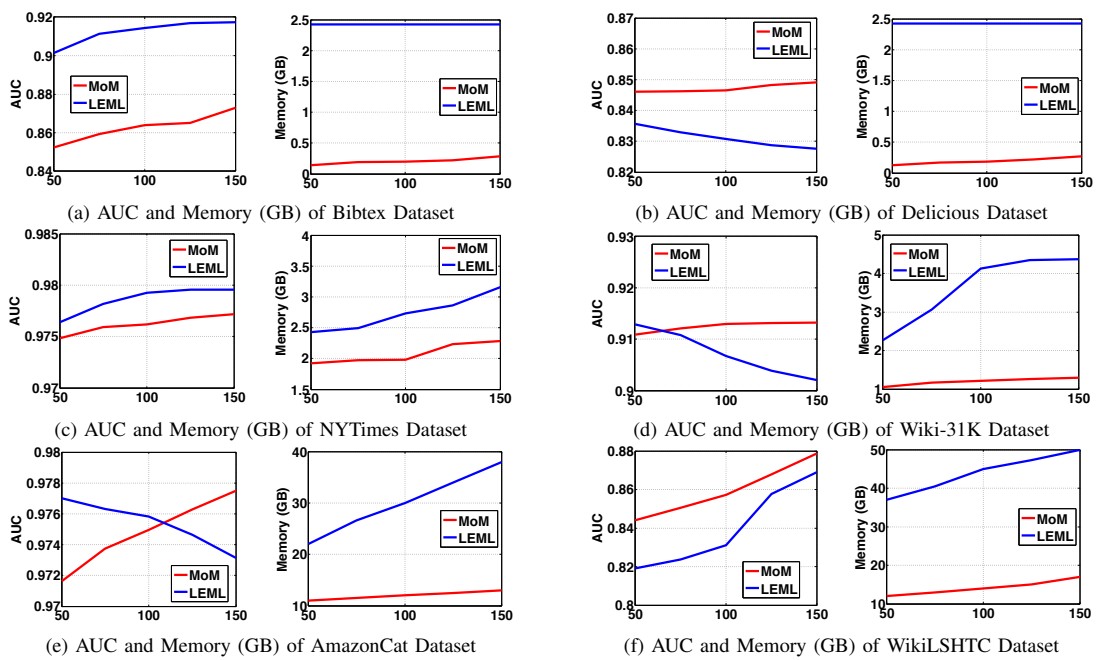

Figure 2: AUC and Memory vs. Latent Dimensionality ($K$)

| Dataset | LEML | MoM | Speed-up ($\times$) |
|---|---|---|---|
| Bibtex | 160s. | 300s. | 0.53 |
| Delicious | 60s. | 150s. | 0.4 |
| NYtimes | 1 hour | 6 min. | 10 |
| Wiki-31K | 3 hr. 40 min. | 15 min. | 15 |
| AmazonCat | 13 hr. [†] | 1 hr. 15 min. | 10 |
| WikiLSHTC | >2 days [†] | 3 hr. | 16 |

† Runtime on i2.4xlarge instance of Amazon EC2

Table 3: Training Time

We computed AUC for every test documents and performed a macro-averaging across the documents, and repeated the experiments for $K = \{50, 75, 100, 125, 150\}$ (Figure 2). Both LEML and Method of Moments perform very similarly, but the memory footprint of MoM is significantly less than LEML. MoM takes longer to finish for the small datasets since tensor factorisation takes much more time compared to the LEML iterations. However, as the size of the datasets grows, the LEML iterations become more and more costly. For the medium and large datasets, MoM takes a fraction of the time taken by LEML. For WikiLSHTC, LEML takes more than two days to finish, while MoM finishes within a few hours. The training times of LEML and MoM for different datasets are listed in Table 3.

## 6 CONCLUSION

Here we propose a latent variable model for multi-label learning in large-scale text corpus and show how to extract the model parameters based on the spectral decomposition of the probability moments of the words and the labels. Our method gives similar performance in comparison with state-of-art algorithms like LEML while taking a fraction of time and memory for the medium and the large datasets. Our method takes only three passes through the training dataset to extract all

the parameters, which contributes to its superior time performance. Also, the memory requirement of our method is nominal when compared to the existing algorithms, and it scales to large datasets like WikiLSHTC containing millions of Wikipedia articles just on a single node with 16GB of memory. Lastly, since our method consists of only linear algebraic operations, it is embarrassingly parallel and can easily be scaled up in any parallel ecosystem using linear algebra libraries. In our implementation, we used Matlab's linear algebra library based on LAPACK/ARPACK, although we did not incorporate any parallelisation.

LEML has a bound of $O\big(1/\sqrt{N}\big)$ on the training loss Yu et al. (2014), whereas our method (MoM) has a convergence bound of $O\big(1/\sqrt{N}\big)$ on the model parameters w.r.t. the number of training instances $N$. However, when we compute AUC on the test dataset, the AUC of LEML decreases with the latent dimensionality($K$) for three datasets including AmazonCat with more than a million of training instances. It shows the possibility of overfitting in LEML. MoM, on the other hand, is not an optimisation algorithm, and the parameters are extracted through spectral decomposition of matrices and tensors, rather than optimising any target function. It is not susceptible to over-fitting, which is evident from its performance. But MoM has the requirement of $N \geq \Omega(K^2)$, and will not work when $N < \Theta(K^2)$. However, for smaller text corpora where $N < \Theta(K^2)$ holds, 1-vs-all classifiers are usually sufficient to predict the labels. We need low-rank models for large text corpora where 1-vs-all classifiers fail, and MoM provides a very competitive choice for such cases.

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

## A  MATRIX NORM INEQUALITIES

Here we paraphrase the inequalities regarding the matrix norms from Anandkumar et al. (2014), which we will use further in our final proof. Let the true pairwise probability matrix and the third order probability moment be $M_2 = p(v, v)$ and $M_3 = p(v, v, v)$, where $v$ stands for the words. Let us assume that we select $N$ i.i.d. samples $x_1, \ldots x_N$ from the population, and the estimates of pairwise matrix and third order moment are $\hat{M}_2 = \hat{p}(y, y)$ and $\hat{M}_3 = \hat{p}(y, y, y)$. Let $\varepsilon_{M_2} = ||M_2 - \hat{M}_2||_2$. We use the second order operator norm of the matrices here. Let us assume $\varepsilon_{M_2} \leq \sigma_K(M_2)/2$, where $\sigma_K$ is the $K$th largest eigenvalue of $M_2$. We will derive the conditions which satisfies this later.

If $\Sigma = diag(\sigma_1, \sigma_2 \ldots \sigma_K)$ are the top-K eigenvalues of $M_2$, and $U$ are the corresponding eigenvectors, then the whitening matrix $W = U\Sigma^{-1/2}$. Also, $W^\top M_2 W = I_{K \times K}$. Then,

$$||W||_2 = \sqrt{\max \mathrm{eig}(W^\top W)} = \sqrt{\max \mathrm{eig}(\Sigma^{-1})} = \frac{1}{\sqrt{\sigma_K(M_2)}}$$

Similarly, if $W^\dagger = W(W^\top W)^{-1}$, then $W^\dagger = W\Sigma = U\Sigma^{1/2}$. Therefore,

$$||W^\dagger||_2 = \sqrt{\max \mathrm{eig}(\Sigma)} = \sqrt{\sigma_1(M_2)} \tag{21}$$

Let $\hat{W}$ be the whitening matrix for $\hat{M}_2$, i.e., $\hat{W}^\top \hat{M}_2 \hat{W} = I_{K \times K}$. Then by Weyl's inequality, $\sigma_k(M_2) - \sigma_k(\hat{M}_2) \leq ||M_2 - \hat{M}_2||, \forall k = 1, 2 \ldots K$.

Therefore,

$$||\hat{W}||_2^2 = \frac{1}{\sigma_K(\hat{M}_2)} \leq \frac{1}{\sigma_K(M_2) - ||M_2 - \hat{M}_2||} \leq \frac{2}{\sigma_K(M_2)} \tag{22}$$

Also, by Weyl's Theorem,

$$||\hat{W}^\dagger||_2^2 = \sigma_1(\hat{M}_2) \leq \sigma_1(M_2) + \varepsilon_{M_2} \leq 1.5\sigma_1(M_2) \implies ||\hat{W}^\dagger||_2 \leq \sqrt{1.5\sigma_1(M_2)} \leq 1.5\sqrt{\sigma_1(M_2)} \tag{23}$$

Let $U^*$ be the eigenvectors of $\hat{W}M_2\hat{W}$, and $\Lambda$ be the corresponding eigenvalues. Then we can write, $\hat{W}M_2\hat{W} = U^*\Lambda U^{*\top}$. Then $W = \hat{W}U^*\Lambda^{-1/2}U^{*\top}$ whitens $M_2$, i.e., $W^\top M_2 W = I$. Therefore,

$$
\begin{aligned}
||I - \Lambda||_2 &= ||I - U^*\Lambda U^{*\top}||_2 \\
&= ||I - \hat{W}M_2\hat{W}||_2 \\
&= ||\hat{W}\hat{M}_2\hat{W} - \hat{W}M_2\hat{W}||_2 \\
&\leq ||\hat{W}||_2^2||M_2 - \hat{M}_2|| \\
&\leq \frac{2}{\sigma_K(M_2)}\varepsilon_{M_2}
\end{aligned} \tag{24}
$$

$$
\begin{aligned}
\varepsilon_W &= ||W - \hat{W}|| \\
&= ||W - WU^*\Lambda^{1/2}U^{*\top}||_2 \\
&= ||W||_2||I - U^*\Lambda^{1/2}U^{*\top}||_2 \\
&= ||W||_2||I - \Lambda^{1/2}||_2 \\
&\leq ||W||_2||I - \Lambda||_2 \\
&\leq \frac{2}{\sigma_K(M_2)^{3/2}}\varepsilon_{M2}
\end{aligned} \tag{25}
$$

$$
\begin{aligned}
\varepsilon_{W^\dagger} &= ||W^\dagger - \hat{W}^\dagger||_2 \\
&= ||\hat{W}^\dagger U^*\Lambda^{1/2}U^{*\top} - \hat{W}^\dagger||_2 \\
&= ||\hat{W}^\dagger||_2||I - U^*\Lambda^{1/2}U^{*\top}||_2 \\
&\leq ||\hat{W}^\dagger||_2||I - \Lambda||_2 \leq \frac{2\sqrt{\sigma_1(M_2)}}{\sigma_K(M_2)}\varepsilon_{M_2}
\end{aligned} \tag{26}
$$

## B  TENSOR NORM INEQUALITIES

Let us define the second order operator norm of a tensor $T \in \mathbb{R}^{D' \times D \times D}$ with $D, D' \in \mathbb{Z}^+$ as,

$$||T||_2 = \sup_u \{||T(\cdot, u, u)|| : u \in \mathbb{R}^D \& ||u|| = 1\} \tag{27}$$

**Lemma 1.** *For a tensor $T \in \mathbb{R}^{D' \times D \times D}$, $||T||_2 \leq ||T||_F$, where $||T||_F$ is the Frobenius norm defined as,*

$$||T||_F = \sqrt{\sum_{i,j,k}(T_{i,j,k})^2}$$

*Proof.* The tensor $T$ can be unfolded as an array of $D'$ matrices each of size $D \times D$, as $T = [T_1, T_2 \ldots T_{D'}]$. Then,

$$T(\cdot, u, u) = \left[u^\top T_1 u, u^\top T_2 u, \ldots u^\top T_{D'} u\right] \in \mathbb{R}^{D'}$$

Therefore,

$$
\begin{aligned}
||T||_2^2 &= \sup_{||u||=1} \left|\left| [u^\top T_1 u, u^\top T_2 u, \dots u^\top T_{D'} u] \right|\right|^2 \\
&= \sup_{||u||=1} \left( |u^\top T_1 u|^2 + |u^\top T_2 u|^2 + \dots + |u^\top T_{D'} u|^2 \right) \\
&\leq \sup_{||u||=1} |u^\top T_1 u|^2 + \sup_{||u||=1} |u^\top T_2 u|^2 + \dots + \sup_{||u||=1} |u^\top T_{D'} u|^2
\end{aligned}
\tag{28}
$$

Let us assume that the singular value decomposition of $T_d$ has the form ($d \in [D']$),

$$
T_d = \sigma_1 \psi_1 \phi_1^\top + \sigma_2 \psi_2 \phi_2^\top + \dots + \sigma_D \psi_D \phi_D^\top
$$

where $\{\psi_i\}_{i=1}^D$ are the left singular vectors, $\{\phi_i\}_{i=1}^D$ are the right singular vectors and $\{\sigma_i\}_{i=1}^D$ are the singular values.

Each of $[\psi_1, \psi_2 \dots \psi_D]$ and $[\phi_1, \phi_2 \dots \phi_D]$ forms a spanning set in $\mathbb{R}^D$, and any vector $u \in \mathbb{R}^D$ can be spanned using both $\psi_{1:D}$ and $\phi_{1:D}$ as the basis.

Let $u = \alpha_1 \psi_1 + \alpha_2 \psi_2 + \dots + \alpha_D \psi_D = \beta_1 \phi_1 + \beta_2 \phi_2 + \dots + \beta_D \phi_D$, where $\alpha_{1:D}$ and $\beta_{1:D}$ are scalars.

Since $||u|| = \sqrt{<u,u>} = 1$,

$$
||\alpha|| = \sqrt{\alpha_1^2 + \alpha_2^2 + \dots \alpha_D^2} = 1 \text{ and } ||\beta|| = \sqrt{\beta_1^2 + \beta_2^2 + \dots \beta_D^2} = 1
$$

Therefore,

$$
\begin{aligned}
|u^\top T_d u| &= |\alpha_1 \sigma_1 \beta_1 + \alpha_2 \sigma_2 \beta_2 \dots \alpha_D \sigma_D \beta_D| \\
&\leq \sqrt{\sigma_1^2 + \sigma_2^2 + \dots + \sigma_D^2} \sqrt{\alpha_1^2 \beta_1^2 + \alpha_2^2 \beta_2^2 \dots \alpha_D^2 \beta_D^2} \quad \text{(Cauchy-Schwartz Inequality)} \\
&\leq \sqrt{\sigma_1^2 + \sigma_2^2 + \dots + \sigma_D^2} \sqrt{\alpha_1^2 + \alpha_2^2 \dots + \alpha_D^2} \sqrt{\beta_1^2 + \beta_2^2 \dots + \beta_D^2} \\
&= \sqrt{\sigma_1^2 + \sigma_2^2 + \dots + \sigma_D^2} \\
&= ||T_d||_F
\end{aligned}
$$

Equality holds when $T_d$ is of rank 1. Since this holds for any vector $u \in \mathbb{R}^D$ such that $||u|| = 1$, $\sup_{||u||=1} |u^\top T_d u| \leq ||T_d||_F$ for $d \in [D']$. Therefore, from Equation 28,

$$
||T||_2^2 \leq \left( ||T_1||_F^2 + ||T_2||_F^2 + \dots + ||T_{D'}||_F^2 \right) = ||T||_F^2 \implies ||T||_2 \leq ||T||_F
$$

$\square$

**Lemma 2.** *(Robust Power Method from (Anandkumar et al., 2014)) If $\hat{T} = T + E \in \mathbb{R}^{K \times K \times K}$, where $T$ is an symmetric tensor with orthogonal decomposition $T = \sum_{k=1}^K \lambda_k u_k \otimes u_k \otimes u_k$ with each $\lambda_k > 0$, and $E$ has operator norm $||E||_2 \leq \epsilon$. Let $\lambda_{\min} = \min_{k=1}^K \{\lambda_k\}$ and $\lambda_{\max} = \max_{k=1}^K \{\lambda_k\}$. Let there exist constants $c_1, c_2$ such that $\epsilon \leq c_1 \cdot (\lambda_{\min}/K)$, and $N \geq c_2 (\log K + \log\log(\lambda_{\max}/\epsilon))$. Then if Algorithm 1 in (Anandkumar et al., 2014) is called for $K$ times, with $L = poly(K) \log(1/\eta)$ restarts each time for some $\eta \in (0,1)$, then with probability at least $1 - \eta$, there exists a permutation $\Pi$ on $[K]$, such that,*

$$
||u_{\Pi(k)} - \hat{u}_k|| \leq 8 \frac{\epsilon}{\lambda_{\Pi(k)}}, \ |\lambda_k - \lambda_{\Pi(k)}| \leq 5\epsilon \ \forall k \in [K]
\tag{29}
$$

Since $\epsilon \leq c_1 \cdot (\lambda_{\min}/K)$ and $\lambda_k = \frac{1}{\sqrt{\pi_k}}, \forall k \in [K]$, we need

$$
N \geq c_2 \left( \log K + \log\log \left( \frac{K \lambda_{\max}}{c_1 \lambda_{\min}} \right) \right) = c_2 \left( \log K + \log\log \left( \frac{K}{c_1} \sqrt{\frac{\pi_{max}}{\pi_{min}}} \right) \right)
\tag{30}
$$

This contributes in the first lower bound ($n_1$) of $N$ in Theorem 1.

## C    TAIL INEQUALITY

**Lemma 3.** *If we draw $N$ i.i.d. sample documents $x_1, x_2 \ldots x_N$, and probability mass function, pairwise probability and triplet probability of the words $v$ estimated from these $N$ samples are $\hat{p}(v)$, $\hat{p}(v, v)$ and $\hat{p}(v, v, v)$ respectively, whereas the true probabilities are $p(v)$, $p(v, v)$ and $p(v, v, v)$ respectively with $v \in \{v_1, v_2 \ldots v_D\}$ , then with probability at least $1 - \delta$ with $\delta \in (0, 1)$,*

$$||\hat{p}(v) - p(v)||_F \leq \frac{2}{\tilde{d}_{1s}\sqrt{N}} \left( 1 + \sqrt{\frac{\log(1/\delta)}{2}} \right) \tag{31}$$

$$||\hat{p}(v, v) - p(v, v)||_F \leq \frac{2}{\tilde{d}_{2s}\sqrt{N}} \left( 1 + \sqrt{\frac{\log(1/\delta)}{2}} \right) \tag{32}$$

$$||\hat{p}(v, v, v) - p(v, v, v)||_F \leq \frac{2}{\tilde{d}_{3s}\sqrt{N}} \left( 1 + \sqrt{\frac{\log(1/\delta)}{2}} \right) \tag{33}$$

*where, $\tilde{d}_{1s} = \mathbb{E}\left[nnz(x)\right]$, $\tilde{d}_{2s} = \mathbb{E}\left[nnz(x)^2\right]$, $\tilde{d}_{3s} = \mathbb{E}\left[nnz(x)^3\right]$, and $nnz(x)$ is the non-zero entries in the binary vector representing the words in the documents as described in Section 3 (main paper).*

*Proof.* Since the number of distinct words in a document is always bounded, we can assume $||x|| \leq 1 \ \forall x$ without loss of generality. Then from Lemma 7 of supplementary material of Wang & Zhu (2014), with probability at least $1 - \delta$ with $\delta \in (0, 1)$,

$$\left|\left|\hat{\mathbb{E}}[x] - \mathbb{E}[x]\right|\right|_F \leq \frac{2}{\sqrt{N}} \left( 1 + \sqrt{\frac{\log(1/\delta)}{2}} \right) \tag{34}$$

$$\left|\left|\hat{\mathbb{E}}[xx^\top] - \mathbb{E}[xx^\top]\right|\right|_F \leq \frac{2}{\sqrt{N}} \left( 1 + \sqrt{\frac{\log(1/\delta)}{2}} \right) \tag{35}$$

$$\left|\left|\hat{\mathbb{E}}[x \otimes x \otimes x] - \mathbb{E}[x \otimes x \otimes x]\right|\right|_F \leq \frac{2}{\sqrt{N}} \left( 1 + \sqrt{\frac{\log(1/\delta)}{2}} \right) \tag{36}$$

where $\mathbb{E}$ stands for true expectation, and $\hat{\mathbb{E}}$ stands for the expectation estimated from the $N$ samples, i.e.,

$$\hat{\mathbb{E}}[x] = \frac{1}{N} \sum_{i=1}^N x_i = \frac{1}{N} X^\top \mathbf{1}$$

$$\hat{\mathbb{E}}[xx^\top] = \frac{1}{N} \sum_{i=1}^N x_i \otimes x_i = \frac{1}{N} X^\top X$$

$$\hat{\mathbb{E}}[x \otimes x \otimes x] = \frac{1}{N} \sum_{i=1}^N x_i \otimes x_i \otimes x_i = \frac{1}{N} X \otimes X \otimes X$$

Now, since the samples contains binary data, probability of the items can be computed from as

$$p(v) = \frac{\mathbb{E}[x]}{\sum_v \mathbb{E}[x_v]} \tag{37}$$

Similarly,

$$\hat{p}(v) = \frac{\hat{\mathbb{E}}[x]}{\sum_v \hat{\mathbb{E}}[x_v]} \tag{38}$$

Now, since $x$ is a binary vector of dimension $D$, the sum across its dimensions is $\sum_{v=1}^{D} x_v = nnz(x)$, and therefore, $\sum_{v=1}^{D} \mathbb{E}[x_v] = \mathbb{E}[\sum_{v=1}^{D} x_v] = \mathbb{E}[nnz(x)] \approx \hat{\mathbb{E}}[nnz(x)] = \sum_{v=1}^{D} \hat{\mathbb{E}}[x_v]$. Assigning $\tilde{d}_{1s} = \mathbb{E}[nnz(x)]$, from Equation 37 and 38, we get

$$\hat{p}(v) - p(v) = \frac{\hat{\mathbb{E}}[x] - \mathbb{E}[x]}{\tilde{d}_{1s}} \tag{39}$$

Similarly, summing across the respective dimensions, we get $\sum_{v_1=1}^{D} \sum_{v_2=1}^{D} (xx^\top)_{v_1,v_2} =$ $\sum_{v_1=1}^{D} \sum_{v_2=1}^{D} x_{v_1} x_{v_2} = nnz(x)^2$ and $\sum_{v_1=1}^{D} \sum_{v_2=1}^{D} \sum_{v_3=1}^{D} (x \otimes x \otimes x)_{v_1,v_2,v_3} = nnz(x)^3$.

Therefore, $\sum_{v_1=1}^{D} \sum_{v_2=1}^{D} \mathbb{E}[(xx^\top)_{v_1,v_2}] = \mathbb{E}[nnz(x)^2] = \tilde{d}_{2s} \approx \sum_{v_1=1}^{D} \sum_{v_2=1}^{D} \hat{\mathbb{E}}[(xx^\top)_{v_1,v_2}]$, and $\sum_{v_1=1}^{D} \sum_{v_2=1}^{D} \sum_{v_3=1}^{D} \mathbb{E}[(x \otimes x \otimes x)_{v_1,v_2,v_3}] = \mathbb{E}[nnz(x)^3] = \tilde{d}_{3s} \approx \sum_{v_1=1}^{D} \sum_{v_2=1}^{D} \sum_{v_3=1}^{D} \hat{\mathbb{E}}[(x \otimes x \otimes x)_{v_1,v_2,v_3}]$.

From here, we can show that,

$$\hat{p}(v,v) - p(v,v) = \frac{\hat{\mathbb{E}}[xx^\top] - \mathbb{E}[xx^\top]}{\tilde{d}_{2s}}$$

$$\hat{p}(v,v,v) - p(v,v,v) = \frac{\hat{\mathbb{E}}[x \otimes x \otimes x] - \mathbb{E}[x \otimes x \otimes x]}{\tilde{d}_{3s}} \tag{40}$$

Plugging these equations in Equation 34, 35 and 36, we can complete the proof.

$\square$

Also, if $y$ represents the label vector of the documents, since the number of labels per documents is limited, we can assume $||y|| \le 1$ without a loss of generality. Then,

$$\hat{\mathbb{E}}[y \otimes x \otimes x] - \mathbb{E}[y \otimes x \otimes x]$$

$$= \frac{1}{N} \sum_{i=1}^{N} y_i \otimes x_i \otimes x_i - \mathbb{E}[y \otimes x \otimes x]$$

$$= \frac{1}{N} \sum_{i=1}^{N} y_i \otimes x_i \otimes x_i - \frac{1}{N} \sum_{i=1}^{N} y_i \otimes \mathbb{E}[x \otimes x] + \frac{1}{N} \sum_{i=1}^{N} y_i \otimes \mathbb{E}[x \otimes x] - \mathbb{E}[y \otimes x \otimes x]$$

$$= \frac{1}{N} \sum_{i=1}^{N} y_i \otimes \left( x_i \otimes x_i - \mathbb{E}[x \otimes x] \right) + \frac{1}{N} \sum_{i=1}^{N} y_i \otimes \mathbb{E}[x \otimes x] - \mathbb{E}[y \otimes x \otimes x]$$

Therefore,

$$\left|\left| \hat{\mathbb{E}}[y \otimes x \otimes x] - \mathbb{E}[y \otimes x \otimes x] \right|\right|_F \le \left|\left| \hat{\mathbb{E}}[y] \right|\right| \left|\left| \hat{\mathbb{E}}[x \otimes x] - \mathbb{E}[x \otimes x] \right|\right|_F + \left|\left| \hat{\mathbb{E}}[y] - \mathbb{E}[y] \right|\right| \left|\left| \mathbb{E}[x \otimes x] \right|\right|_F$$

$$= \left|\left| \hat{\mathbb{E}}[y] \right|\right| \left|\left| \hat{\mathbb{E}}[xx^\top] - \mathbb{E}[xx^\top] \right|\right|_F + \left|\left| \hat{\mathbb{E}}[y] - \mathbb{E}[y] \right|\right| \left|\left| \mathbb{E}[xx^\top] \right|\right|_F$$

Since $||y|| \le 1$, $||\hat{\mathbb{E}}[y]|| \le 1$. Also, from $||x|| \le 1$, we get $||\mathbb{E}[xx^\top]|| \le ||x||^2 \le 1$. Therefore,

$$\left|\left| \hat{\mathbb{E}}[y \otimes x \otimes x] - \mathbb{E}[y \otimes x \otimes x] \right|\right|_F \le \left|\left| \hat{\mathbb{E}}[xx^\top] - \mathbb{E}[xx^\top] \right|\right|_F + \left|\left| \hat{\mathbb{E}}[y] - \mathbb{E}[y] \right|\right| \tag{41}$$

From Equation 34 and 35,

$$P\left[\left\|\hat{\mathbb{E}}[y] - \mathbb{E}[y]\right\|_F \leq \frac{2}{\sqrt{N}}\left(1 + \sqrt{\frac{\log(1/\delta)}{2}}\right)\right] \geq 1 - \delta$$

$$P\left[\left\|\hat{\mathbb{E}}[xx^\top] - \mathbb{E}[xx^\top]\right\|_F \leq \frac{2}{\sqrt{N}}\left(1 + \sqrt{\frac{\log(1/\delta)}{2}}\right)\right] \geq 1 - \delta$$

Reversing the tail inequalities,

$$P\left[\left\|\hat{\mathbb{E}}[y] - \mathbb{E}[y]\right\|_F \geq \frac{2}{\sqrt{N}}\left(1 + \sqrt{\frac{\log(1/\delta)}{2}}\right)\right] \leq \delta$$

$$P\left[\left\|\hat{\mathbb{E}}[xx^\top] - \mathbb{E}[xx^\top]\right\|_F \geq \frac{2}{\sqrt{N}}\left(1 + \sqrt{\frac{\log(1/\delta)}{2}}\right)\right] \leq \delta$$

If $\mathcal{E}_1$ and $\mathcal{E}_2$ are two events,

$$P(\mathcal{E}_1 \cap \mathcal{E}_2) = P(\mathcal{E}_1) + P(\mathcal{E}_2) - P(\mathcal{E}_1 \cup \mathcal{E}_2)$$
$$\leq P(\mathcal{E}_1) + P(\mathcal{E}_2)$$

Therefore,

$$P\left[\left\|\hat{\mathbb{E}}[y] - \mathbb{E}[y]\right\|_F + \left\|\hat{\mathbb{E}}[xx^\top] - \mathbb{E}[xx^\top]\right\|_F \geq \frac{4}{\sqrt{N}}\left(1 + \sqrt{\frac{\log(1/\delta)}{2}}\right)\right] \leq 2\delta$$

Or,

$$P\left[\left\|\hat{\mathbb{E}}[y] - \mathbb{E}[y]\right\|_F + \left\|\hat{\mathbb{E}}[xx^\top] - \mathbb{E}[xx^\top]\right\|_F \leq \frac{4}{\sqrt{N}}\left(1 + \sqrt{\frac{\log(1/\delta)}{2}}\right)\right] \geq 1 - 2\delta$$

From Equation 41,

$$P\left[\left\|\hat{\mathbb{E}}[y \otimes x \otimes x] - \mathbb{E}[y \otimes x \otimes x]\right\|_F \leq \frac{4}{\sqrt{N}}\left(1 + \sqrt{\frac{\log(1/\delta)}{2}}\right)\right] \geq 1 - 2\delta$$

Replacing $\delta$ by $\delta/2$,

$$P\left[\left\|\hat{\mathbb{E}}[y \otimes x \otimes x] - \mathbb{E}[y \otimes x \otimes x]\right\|_F \leq \frac{4}{\sqrt{N}}\left(1 + \sqrt{\frac{\log(2/\delta)}{2}}\right)\right] \geq 1 - \delta$$

Similar to previous cases $\sum_{l=1}^{L} \sum_{v_1=1}^{D} \sum_{v_2=1}^{D} (y \otimes x \otimes x)_{l,v_1,v_2} = nnz(y)nnz(x)^2$, and therefore, $\sum_{l=1}^{L} \sum_{v_1=1}^{D} \sum_{v_2=1}^{D} \mathbb{E}[(y \otimes x \otimes x)_{l,v_1,v_2}] = \mathbb{E}[nnz(y)nnz(x)^2] \approx \hat{\mathbb{E}}[nnz(y)nnz(x)^2]$. Assigning $\tilde{d}_{ls} = \mathbb{E}[nnz(y)nnz(x)^2]$, we can show that,

$$\hat{p}(l, v, v) - p(l, v, v) = \frac{\hat{\mathbb{E}}[y \otimes x \otimes x] - \mathbb{E}[y \otimes x \otimes x]}{\tilde{d}_{ls}} \tag{42}$$

where $l$ stands for the labels ($l \in \{l_1, l_2 \ldots l_L\}$). Therefore, with probability at least $1 - \delta$,

$$\|\hat{p}(l, v, v) - p(l, v, v)\|_F \leq \frac{4}{\tilde{d}_{ls}\sqrt{N}}\left(1 + \sqrt{\frac{\log(2/\delta)}{2}}\right) \tag{43}$$

## D  COMPLETING THE PROOF

Assigning $\varepsilon_1 = \left(1 + \sqrt{\frac{\log(1/\delta)}{2}}\right)$ in the inequalities of Lemma 3, we get

$$\varepsilon_{M_2} = ||M_2 - \hat{M}_2||_2 = ||\hat{p}(v,v) - p(v,v)||_2 \leq ||\hat{p}(v,v) - p(v,v)||_F \leq \frac{2\varepsilon_1}{\tilde{d}_{2s}\sqrt{N}}, \text{ and}$$

$$\varepsilon_{M_3} = ||M_3 - \hat{M}_3||_2 = ||\hat{p}(v,v,v) - p(v,v,v)||_2 \leq ||\hat{p}(v,v,v) - p(v,v,v)||_F \leq \frac{2\varepsilon_1}{\tilde{d}_{3s}\sqrt{N}}$$

since operator norm is smaller than Frobenius norm for both matrices and tensors (Lemma 1).

Therefore, to satisfy $\varepsilon_{M_2} \leq \sigma_K(M_2)/2$, we need $N \geq \Omega\left(\left(\frac{\varepsilon_1}{\tilde{d}_{2s}\sigma_K(M_2)}\right)^2\right)$. This contributes in the second lower bound ($n_2$) of $N$ in Theorem 1.

Similarly, assigning $\varepsilon_2 = \left(1 + \sqrt{\frac{\log(2/\delta)}{2}}\right)$ in Equation 43, we get

$$\varepsilon_{M_L} = ||M_L - \hat{M}_L||_2 = ||\hat{p}(l,v,v) - p(l,v,v)||_2 \leq ||\hat{p}(l,v,v) - p(l,v,v)||_F \leq \frac{4\varepsilon_2}{\tilde{d}_{ls}\sqrt{N}}$$

From Appendix B in (?),

$$\begin{aligned}
\varepsilon_{tw} &= ||M_3(W,W,W) - \hat{M}_3(\hat{W},\hat{W},\hat{W})||_2 \\
&\leq ||M_3||_2 \left(||\hat{W}||_2^2 + ||\hat{W}||_2||W||_2 + ||W||_2^2\right)\varepsilon_W + ||\hat{W}||^3 \varepsilon_{M_3} \\
&\leq ||M_3||_2 \frac{(2 + \sqrt{2} + 1)}{\sigma_K(M_2)}\varepsilon_W + \frac{2\sqrt{2}}{\sigma_K(M_2)^{3/2}}\varepsilon_{M_3} \\
&\leq ||M_3||_2 \frac{(3 + \sqrt{2})}{\sigma_K(M_2)} \cdot \frac{2}{\sigma_K(M_2)^{3/2}}\varepsilon_{M2} + \frac{2\sqrt{2}}{\sigma_K(M_2)^{3/2}}\varepsilon_{M_3} \\
&\leq \frac{10||M_3||_2}{\sigma_K(M_2)^{5/2}} \cdot \varepsilon_{M2} + \frac{2\sqrt{2}}{\sigma_K(M_2)^{3/2}}\varepsilon_{M_3} \\
&\leq \left(\frac{10}{\tilde{d}_{2s}\sigma_K(M_2)^{5/2}} + \frac{2\sqrt{2}}{\tilde{d}_{3s}\sigma_K(M_2)^{3/2}}\right)\frac{2\varepsilon_1}{\sqrt{N}}
\end{aligned} \tag{44}$$

From Lemma 1, $||M_3||_2 \leq ||M_3||_F \leq 1$, because $M_3$ is a tensor with individual elements as probabilities, with the sum of all elements being 1.

From Lemma 2, $\epsilon \leq c_1 \cdot (\lambda_{\min}/K)$, and we can assign $\epsilon$ as the upper bound of $\varepsilon_{tw}$. To satisfy this, we need

$$\left(\frac{10}{\tilde{d}_{2s}\sigma_K(M_2)^{5/2}} + \frac{2\sqrt{2}}{\tilde{d}_{3s}\sigma_K(M_2)^{3/2}}\right)\frac{2\varepsilon}{\sqrt{N}} \leq c_1 \frac{\lambda_{\min}}{K}, \text{ or,}$$

$$\left(\frac{10}{\tilde{d}_{2s}\sigma_K(M_2)^{5/2}} + \frac{2\sqrt{2}}{\tilde{d}_{3s}\sigma_K(M_2)^{3/2}}\right)\frac{2\varepsilon}{\sqrt{N}} \leq c_1 \frac{1}{K\sqrt{\pi_{\max}}}$$

Since $\pi_{\max} \leq 1$, we need $N \geq \Omega\left(K^2\left(\frac{10}{\tilde{d}_{2s}\sigma_K(M_2)^{5/2}} + \frac{2\sqrt{2}}{\tilde{d}_{3s}\sigma_K(M_2)^{3/2}}\right)^2 \varepsilon^2\right)$. This contributes to $n_3$ in Theorem 1.

Here, we will derive the convergence bounds for the parameters. Since $\mu_k = W^\dagger u_k$ (Algorithm 1 in main paper), with probability at least $1 - \delta$,

$$
\begin{aligned}
||\mu_k - \hat{\mu}_k|| \\
&= ||W^\dagger u_k - \hat{W}^\dagger \hat{u}_k|| \\
&= ||W^\dagger u_k - W^\dagger \hat{u}_k + W^\dagger \hat{u}_k - \hat{W}^\dagger \hat{u}_k|| \\
&\leq ||W^\dagger||_2 ||u_k - \hat{u}_k|| + ||W^\dagger - \hat{W}^\dagger||_2 ||\hat{u}_k|| \\
&\leq ||W^\dagger||_2 \frac{8\epsilon}{\lambda_k} + \varepsilon_{W^\dagger} \\
&\leq 8\sqrt{\sigma_1(M_2)}\epsilon + \frac{2\sqrt{\sigma_1(M_2)}}{\sigma_K(M_2)}\varepsilon_{M_2}
\end{aligned}
\tag{45}
$$

where $||u_k - \hat{u}_k|| \leq 8\epsilon/\lambda_k$ from Lemma 2, and $||\hat{u}_k|| = 1$. Since $1/\lambda_k = \sqrt{\pi_k} \leq 1$, assigning $\epsilon$ as the upper bound of $\varepsilon_{tw}$ in Equation 44, we can say that with probability at least $1 - \delta$,

$$
\begin{aligned}
||\mu_k - \hat{\mu}_k|| &\leq 8\sqrt{\sigma_1(M_2)}\left(\frac{10}{\tilde{d}_{2s}\sigma_K(M_2)^{5/2}} + \frac{2\sqrt{2}}{\tilde{d}_{3s}\sigma_K(M_2)^{3/2}}\right)\frac{2\varepsilon_1}{\sqrt{N}} + \frac{2\sqrt{\sigma_1(M_2)}}{\sigma_K(M_2)}\frac{2\varepsilon_1}{\tilde{d}_{2s}\sqrt{N}} \\
&\leq \left(\frac{160\sqrt{\sigma_1(M_2)}}{\tilde{d}_{2s}\sigma_K(M_2)^{5/2}} + \frac{32\sqrt{2\sigma_1(M_2)}}{\tilde{d}_{3s}\sigma_K(M_2)^{3/2}} + \frac{4\sqrt{\sigma_1(M_2)}}{\tilde{d}_{2s}\sigma_K(M_2)}\right)\frac{\varepsilon_1}{\sqrt{N}}
\end{aligned}
\tag{46}
$$

Also,

$$
\begin{aligned}
|\pi_k - \hat{\pi}_k| &= \left|\frac{1}{\lambda_k^2} - \frac{1}{\hat{\lambda}_k^2}\right| = \left|\frac{(\lambda_k + \hat{\lambda}_k)(\lambda_k - \hat{\lambda}_k)}{\lambda_k^2 \hat{\lambda}_k^2}\right| = \left|\sqrt{\pi_k \hat{\pi}_k}\left(\sqrt{\pi_k} + \sqrt{\hat{\pi}_k}\right)(\lambda_k - \hat{\lambda}_k)\right| \\
&\leq 2|\lambda_k - \hat{\lambda}_k| \leq 10\epsilon
\end{aligned}
\tag{47}
$$

since $|\lambda_k - \hat{\lambda}_k| \leq 5\epsilon$ from Lemma 2. Therefore, with probability at least $1 - \delta$, we get

$$
|\pi_k - \hat{\pi}_k| \leq \left(\frac{200}{\sigma_K(M_2)^{5/2}} + \frac{40\sqrt{2}}{\sigma_K(M_2)^{3/2}}\right)\frac{\varepsilon_1}{\tilde{d}_{3s}\sqrt{N}}
\tag{48}
$$

where $\varepsilon = \left(1 + \sqrt{\frac{\log(1/\delta)}{2}}\right)$.

Further, since $\gamma = u_k^\top M_L(W, W)u_k$,

$$
\begin{aligned}
||\gamma_k - \hat{\gamma}_k|| \\
&= ||u_k^\top M_L(W, W)u_k - \hat{u}_k^\top \hat{M}_L(\hat{W}, \hat{W})\hat{u}_k|| \\
&\leq ||u_k^\top M_L(W, W)u_k - \hat{u}_k^\top M_L(W, W)\hat{u}_k|| + ||\hat{u}_k^\top M_L(W, W)\hat{u}_k - \hat{u}_k^\top \hat{M}_L(\hat{W}, \hat{W})\hat{u}_k|| \\
&\leq ||u_k^\top M_L(W, W)u_k - \hat{u}_k^\top M_L(W, W)u_k + \hat{u}_k^\top M_L(W, W)u_k - \hat{u}_k^\top M_L(W, W)\hat{u}_k|| + ||\hat{u}_k||^2 ||M_L(W, W) - \hat{M}_L(\hat{W}, \hat{W}) \\
&\leq ||u_k - \hat{u}_k||||M_L(W, W)||_2 ||u_k|| + ||u_k - \hat{u}_k||||M_L(W, W)||_2 ||\hat{u}_k|| + ||\hat{u}_k||^2 ||M_L(W, W) - \hat{M}_L(\hat{W}, \hat{W})||_2 \\
&= 2||u_k - \hat{u}_k||||M_L(W, W)||_2 + ||M_L(W, W) - \hat{M}_L(\hat{W}, \hat{W})||_2 \\
&\leq 2||u_k - \hat{u}_k||||W||^2 ||M_L||_2 + ||M_L(W, W) - \hat{M}_L(\hat{W}, \hat{W})||_2 \\
&\leq 2||u_k - \hat{u}_k||||W||^2 + ||M_L(W, W) - \hat{M}_L(\hat{W}, \hat{W})||_2
\end{aligned}
\tag{49}
$$

since $||u_k|| = ||\hat{u}_k|| = 1$. Also, from Lemma 1, $||M_L||_2 \leq ||M_L||_F \leq 1$, since $||M_L||_2$ is a tensor with the sum of its elements as 1.

Now,

$$
\begin{aligned}
&\left|\left|M_L(W,W) - \hat{M}_L(\hat{W},\hat{W})\right|\right|_2 \\
&= \left|\left|M_L(W,W) - M_L(\hat{W},\hat{W}) + M_L(\hat{W},\hat{W}) - \hat{M}_L(\hat{W},\hat{W})\right|\right|_2 \\
&\leq \left|\left|M_L(W,W) - M_L(\hat{W},\hat{W})\right|\right|_2 + \left|\left|\hat{W}\right|\right|^2 \left|\left|M_L - \hat{M}_L\right|\right|_2 \\
&= \left|\left|M_L(W,W) - M_L(W,\hat{W}) + M_L(W,\hat{W}) - M_L(\hat{W},\hat{W})\right|\right|_2 + \left|\left|\hat{W}\right|\right|^2 \left|\left|M_L - \hat{M}_L\right|\right|_2 \\
&\leq \left|\left|M_L(W,W) - M_L(W,\hat{W})\right|\right|_2 + \left|\left|M_L(W,\hat{W}) - M_L(\hat{W},\hat{W})\right|\right|_2 + \left|\left|\hat{W}\right|\right|^2 \left|\left|M_L - \hat{M}_L\right|\right|_2 \\
&\leq \left|\left|W\right|\right|\left|\left|W - \hat{W}\right|\right|\left|\left|M_L\right|\right|_2 + \left|\left|\hat{W}\right|\right|\left|\left|W - \hat{W}\right|\right|\left|\left|M_L\right|\right|_2 + \left|\left|\hat{W}\right|\right|^2 \left|\left|M_L - \hat{M}_L\right|\right|_2 \\
&= (\left|\left|W\right|\right| + \left|\left|\hat{W}\right|\right|)\left|\left|M_L\right|\right|_2 \varepsilon_W + \left|\left|\hat{W}\right|\right|^2 \varepsilon_{M_L} \\
&\leq (\left|\left|W\right|\right| + \left|\left|\hat{W}\right|\right|)\varepsilon_W + \left|\left|\hat{W}\right|\right|^2 \varepsilon_{M_L} \qquad\qquad (50)
\end{aligned}
$$

Therefore,

$$
\begin{aligned}
&\left|\left|\gamma_k - \hat{\gamma_k}\right|\right| \\
&\leq 2\left|\left|u_k - \hat{u}_k\right|\right|\left|\left|W\right|\right|^2 + (\left|\left|W\right|\right| + \left|\left|\hat{W}\right|\right|)\varepsilon_W + \left|\left|\hat{W}\right|\right|^2 \varepsilon_{M_L} \\
&\leq 16\frac{\epsilon}{\lambda_k}\left|\left|W\right|\right|^2 + (\left|\left|W\right|\right| + \left|\left|\hat{W}\right|\right|)\varepsilon_W + \left|\left|\hat{W}\right|\right|^2 \varepsilon_{M_L} \\
&\leq 16\frac{\epsilon}{\sigma_K(M_2)} + (\left|\left|W\right|\right| + \left|\left|\hat{W}\right|\right|)\varepsilon_W + \left|\left|\hat{W}\right|\right|^2 \varepsilon_{M_L}
\end{aligned}
$$

Since $1/\lambda_k = \sqrt{\pi_k} \leq 1$. Assigning $\epsilon$ as the upper limit of $\varepsilon_{tw}$ in Equation 44, with probability $1 - \delta$,

$$
\begin{aligned}
&\left|\left|\gamma_k - \hat{\gamma_k}\right|\right| \\
&\leq \frac{16}{\sigma_K(M_2)}\left(\frac{10}{\tilde{d}_{2s}\sigma_K(M_2)^{5/2}} + \frac{2\sqrt{2}}{\tilde{d}_{3s}\sigma_K(M_2)^{3/2}}\right)\frac{2\varepsilon_1}{\sqrt{N}} + \frac{1+\sqrt{2}}{\sqrt{\sigma_K(M_2)}}\frac{2}{\sigma_K(M_2)^{3/2}}\frac{2\varepsilon_1}{\sqrt{N}\tilde{d}_{2s}} + \frac{2}{\sigma_K(M_2)}\frac{4\varepsilon_2}{\sqrt{N}\tilde{d}_{ls}} \\
&\leq \left(\frac{80}{\tilde{d}_{2s}\sigma_K(M_2)^{7/2}} + \frac{16\sqrt{2}}{\tilde{d}_{3s}\sigma_K(M_2)^{5/2}} + \frac{1+\sqrt{2}}{\tilde{d}_{2s}\sigma_K(M_2)^2}\right)\frac{4\varepsilon_1}{\sqrt{N}} + \frac{8\varepsilon_2}{\tilde{d}_{ls}\sigma_K(M_2)\sqrt{N}}
\end{aligned}
$$

where, $\varepsilon_1 = \left(1 + \sqrt{\frac{\log(1/\delta)}{2}}\right)$ and $\varepsilon_2 = \left(1 + \sqrt{\frac{\log(2/\delta)}{2}}\right)$. This completes the proof of Theorem 1.

