# OpenReview forum: "Multi-label Learning for Large Text Corpora using Latent Variable Model with Provable Gurantees"
_ICLR.cc/2018/Conference — Reject_

### Official Review · AnonReviewer2 · 2017-11-26
**This paper uses spectral algorithm to learn a latent variable model for multi-label prediction. However the model is very simplistic and the techniques are not very novel.**

**Rating:** 4
**Confidence:** 5

**Review:**

This paper studies the problem of multi-label learning for text copora. The paper proposed a latent variable model for the documents and their labels, and used spectral algorithms to provably learn the parameters.

The model is fairly simplistic: the topic can be one of k topics (pure topic model), based on the topic, there is a probability distribution over documents, and a probabilistic distribution over labels. The model between document and topic is very similar to previous pure topic models (see more discussions below), and because it is a pure topic, the label is just modeled by a conditional distribution.

The paper tried to stress that the model is different from Anandkumar et al. because the use of "expectations vs. probabilities", but that is only different by a normalization factor. The model defined here is also very strange, especially Equation (2) is not really consistent with Equation (7).

Just to elaborate: in equation (2), the probability of a document is related to the set of distinct words, so it does not distinguish between documents where a word appear multiple times or only once. This is different from the standard bag-of-words model where words are sampled independently and word counts do matter. However, in the calculation before Equation (7), it was trying to compute the probability that a pair of words are equal to v_i and v_j, and it assumed words w_1 and w_2 are independent and both of them satisfy the conditional distribution P[v_i|h = k], this is back to the standard bag-of-words model. To see why these models are different, if it is the model of (2), and we look at only distinct words, the diagonal of the matrix P[v_i,v_i] does not really make sense and certainly will not follow Equation (7). Equation (7) and also (9) only works in the standard bag-of-words model that is also used in Anandkumar et al. (the same equations were also proved).

The main novelty in this paper is that it uses the label as a third view of a multi-view model and make use of cross moments. The reviewer feels this alone is not enough contribution.

---

### Official Review · AnonReviewer3 · 2017-11-27
**The paper lacks novelty and the main part is almost the same as Anandkumar et al 2012**

**Rating:** 3
**Confidence:** 4

**Review:**

The paper proposes to learn a latent variable model with spectral algorithm and apply it to multi-label learning. First of all, the connection to multi-label learning is very loose, and the majority of the paper deals with learning the latent variable model. Second, there is almost nothing new in the paper compared to Anandkumar et al 2012, except it uses probability as parameters but not expectations. This difference is trivial since they are only a normalization away. Third, the experiments shows the performance (AUC) compared with other algorithms is significantly worse although the memory consumption may be small.

---

### Official Review · AnonReviewer1 · 2017-11-28
**The clarity of the paper is rather low. The novelty and significance are difficult to assess and seem to be below marginal. Moreover, there are some flaws in the reasoning for the proposed approach. Overall, I don't find this paper to be acceptable for publication in the present form.**

**Rating:** 4
**Confidence:** 5

**Review:**

The paper addresses the problem of multi-label learning for text corpora and proposes to tackle the problem using tensor factorization methods. Some analysis and experimental results for the proposed algorithm are presented.

QUALITY: I find the quality of the results in this paper rather low. The proposed probabilistic model is defined ambiguously. The authors then look at joint probability distributions of co-occurence of two and three words, which gives a matrix and a tensor, respectively. They propose to match these matrix and tensor to their sample estimates and refer to such procedure as the moment matching method, which it is not. They then apply a standard two step technique from the moment matching literature consisting of whitening and orthogonal tensor factorization. However, in their case this does not have much statistical meaning. Indeed, whitening of the covariance matrix is usually justified by the scaling unidentifiability of the problem. In their case, the mathematics works because of the orthogonal unidentifiability of the square root of a matrix. Furthermore, the proposed sample estimators do not actually estimate densities they are dealing with (see, e.g., Eq. (16) and (17)). Their theoretical analysis seems like a straightforward extension of the analysis by Anandkumar, et al. (2012, 2014), however, I find it difficult to assess this analysis due to numerous ambiguities in the problem formulation and method development. This justifies my statement in the beginning of the paragraph.

CLARITY: The paper is not well written and, therefore, is difficult to assess. Many important details are omitted, the formulation of the model is self contradicting, the standard concepts and notations are sometimes abused, some statements are wrong. I provide some examples in the detailed comment below.

ORIGINALITY AND SIGNIFICANCE: The idea to apply tensor factorization approaches to the multi-label learning is novel up to my knowledge and is a pro of the paper. However, I have problems to find other pros in this submission because the clarity is quite low and in the present form there is no novelty in the proposed procedure. Moreover, the authors claim to work with densities, but end up estimating other quantities, which are not guaranteed to have the desirable form. They also emphasize the fact that there is the simplex constraint on the estimated parameters, but this constraint is completely ignored by the algorithm and, in general, won't be satisfied in practice. If think the authors should do some more work before this paper can be published.



DETAILED COMMENTS: Since I am quite critical about the paper, I point out some examples of drawbacks or flaws of this paper:

  - The proposed model (Section 2) is not well defined. In particular, the description in Section 2 is not sufficient to understand the proposed model; the plate diagram in Figure 2 is not consistent with the text. It is not mentioned how at least some conditional distributions behave (e.g., tokens given labels or states). The diagram in Fig. 1 does not help since it isn't consistent with the text (e.g. the elements of labels or states are not conditionally / independent). The model is very close to latent Dirichlet allocation by Blei, et al. (2003), but differences are not discussed.

  - The standard terminology is often abused. For example, the proposed approach is referred to as the method of moments when it is not. In Section 2.1, the authors aim to match joint distributions (not the moments) to their empirical approximations (which are also wrong; see below). The usage of tokes and documents is interchanged without any explanations.

  - The use of the whitening approach is not justified in their setting working with joint distributions of couples and triples and it has no statistical meaning. No explanation is provided. I would definitely not call this whitening.

  - In Section 2.2, the notation is not defined and is different from what is usually used in the literature. For example, Eq. (15) does not make much sense as is. One could guess from the context that they are talking about the eigenvectors of an orthogonal tensor as defined in, e.g. Anandkumar, et al. (2014).

  - In Section 3, the authors emphasize the fact that their parameters are constrained to the probability simplex, but this constraint is not ensured in the proposed algorithm (Alg. 1).

  - Importantly, the estimators of the matrix M_2 and tensor M_3 do not make much sense to me. For example, for estimating M_2 it would be reasonable to average over all word pairs, i.e. something like [M_2]_{ij} = 1/L \sum_{w_k \not = w_l} P(w_k = v_i, w_l = v_j), where L is the number of pairs. This is different from the expression in Eq. (16), which is just a rescaled non-central second moment. Similar issue is true for the order-3 estimator.

  - The factorization procedure does not ensure non-negativity of the obtained parameters and, therefore, the rescaling is not guaranteed to belong to the probability simplex. I could not find any explanations of this issue.

  - I explain good plots in the experimental section, potentially, by the fact that the authors do algorithmically something different from what they aim to do, because the estimators do not estimate the desired entities (i.e. are not consistent). The procedure looks to me quite similar to the procedure for LDA, hence the reasonable results. However, the authors do not justify their proposed method.

---

### Decision · Program_Chairs · 2018-01-29
**ICLR 2018 Conference Acceptance Decision**

**Decision:**

Reject

**Comment:**

There is overall consensus about the paper's lack of novelty and clarity.  Reviewer 1 has detailed comments that can be used to strengthen the paper.  Reviewer 3 suggests that this paper is very close to Anandkumar et al 2012, and it is not clear where the novelty lies.  Addressing these concerns of the reviewers will make the paper more acceptable to future venues.